

# Diet, exercise and mental-wellbeing of healthcare professionals (doctors, dentists and nurses) in Pakistan

Waqas Ahmad[1], Frances Taggart[2], Muhammad Shoaib Shafique[1], Yumna Muzafar[1], Shehnam Abidi[1], Noor Ghani[1], Zahra Malik[1], Tehmina Zahid[1], Ahmed Waqas[1] and Naila Ghaffar[1]

[1] CMH Lahore Medical College and Institute of Dentistry, Lahore Cantt, Pakistan
[2] Statistics and Epidemiology Unit, Division of Health Sciences, University of Warwick Medical School, Coventry, UK

## ABSTRACT

**Background.** "Health is wealth" is a time tested adage. Health becomes more relevant when it comes to professionals whose job is to provide people with services that maintain an optimum state of mental, physical and social well-being. Healthcare professionals (HCP) differ from general population in regards to the nature of their work, stress, burnout etc. which begs the need to have a robust state of health for the ones who provide it to others. We initiated this study to see if healthcare professionals "practice what they preach others."

**Methods.** We employed a cross-sectional study design with convenience-sampling technique. Questionnaires were administered directly to the three groups of healthcare professionals (Doctors, Dentists and Nurses) across the province Punjab after their consent. 1,319 healthcare professionals took part in the study (response rate of 87.35). Warwick Edinburg Mental Wellbeing Scale (WEMWBS) was used to assess mental wellbeing. USDA Dietary Guidelines-2010 were employed to quantify diet. American Heart Association (AHA) guidelines were employed for the analysis of exercise.

**Results.** A total of 1,190 healthcare professionals formed the final sample with doctors and nurses forming the major proportion. Out of 1,190 participants only *one* healthcare professional was found to eat according to USDA Dietary Guidelines; others ate more of protein group and less of fruits, dairy and vegetable groups. 76% did not perform any exercise. 71.5% worked >48 h/week. More than 50% of healthcare professionals were sleeping <7 h/day. WEMWBS score of the entire sample was $47.97 \pm 9.53$ S.D.

**Conclusion.** Our findings suggest that healthcare professionals do not practice what they preach. Their mental wellbeing, diet and exercise habits are not up to the mark and should be improved to foster the whole healthcare system for individual and community benefits.

Corresponding author
Waqas Ahmad,
waqas_lalamusa@yahoo.com

## INTRODUCTION

Health is defined by W.H.O. as "a state of complete physical, mental and social well-being and not merely the absence of disease or infirmity" (*W.H.O, 1948*). Healthy lifestyle has many advantages: a balanced diet lowers the risk of many cardiovascular diseases (*Gambera, Schneeman & Davis, 1995*); regular exercise helps reduce anxiety and depression, lowers cholesterol level and boosts the immune system (*American Heart Association, 2013*); a good mental wellbeing helps cope with daily stress and work productively for individual and community benefit (*Huppert, 2009*), avoidance of first and second hand smoke protects against many respiratory and cardiovascular diseases (*Ockene & Miller, 1997*).

In recent years, mental wellbeing has come to the limelight due to its role in coping and protection against different mental illnesses (*Huppert, 2009*). It equips an individual with resilience, better physical health, financial and personal security, better relationships with friends and family and improves his quality of life (*Deacon et al., 2009*). Mental wellbeing becomes more important when it comes to healthcare professionals who deal with life and death, work long hours, get little time to spend with family and are exposed to violence, insecurity and illegitimate pressures (*Shiwani, 2009*). They are more susceptible to stress and its negative consequences than general population (*Willcock et al., 2004*; *Schattner, Davidson & Serry, 2004*).

Doctors and other healthcare professionals (nurses, dentists) work in harmony to provide optimum care to patients. They do not just work for a disease free community but one with a healthy lifestyle. They provide advice on how to eat right, how to live long, how to be mentally strong to deal with everyday challenges but the questions arise: do they act on their own advice, do they eat right, sleep well, exercise. Are they in a state of good mental well-being to deal with everyday challenges?

Research has shown that doctors who have a healthy lifestyle are more likely to talk to their patients about it (*Oberg & Frank, 2009*) and patients are more likely to take impression from such doctors (*Frank, Breyan & Elon, 2000*). Healthy healthcare professionals not only perform well but also foster the healthcare system which leads to the ultimate goal of medical practice; better patient care. American (*Frank, 2004*) and Canadian (*Frank & Segura, 2009*) doctors are healthier than general population but such studies are much more scarce in Asia as compared to the rest pf world. We initiated this study to see if healthcare professionals of the sixth most populous country in the world "practice what they preach" by chiefly examining their diet, exercise and mental wellbeing along with other aspects which affect their lives e.g., sleep, BMI etc.

## METHODOLOGY

### Subject group and sampling

Descriptive, cross-sectional study design and convenience-sampling technique (non-probability sampling) was employed. A total of 1,510 healthcare professionals were approached out of which 1,319 took part in the study, response rate of 87.35%. Sample size calculation is based on anticipated effect sizes, however, no apriori power estimates were available for multiple regression analysis employed in present study. The authors followed

the sample size recommendations made by *VanVoorhis & Morgan (2007)*. According to Vanvoorhis and Morgan, regression equations with six or more predictors should have a minimum of 10 participants per predictor variable. Three groups of healthcare professionals were included in the study: doctors, nurses and dentists. 129 responses were discarded due to missing dietary intake; essential demographics like age, gender, profession etc. all together and more than 3 missing responses in Warwick Edinburg Mental Wellbeing Scale (WEMWBS) thus making the final sample $N = 1,190$. *Junior doctors* were defined as those doctors who were $\leqslant 30$ years old and in their early years of training while *senior doctors* were defined as all those doctors who were $>30$ years of age. On the basis of monthly income, the following groups were created: Low Income ($\leqslant$PKR. 8,500), Lower-Middle Income ($\geqslant$PKR. 8,501 and $\leqslant$PKR. 33,000), Upper-Middle Income ($\geqslant$PKR. 33,001 and $\leqslant$PKR. 102,000) High-Income ($>$PKR. 102,001). Occupational stressors are defined as all those stresses which are chiefly present in a work place environment (Hospital, clinics etc.). Following stressors were asked about Long Working Hours, Patient Overload, Uncertain Future, Insufficient Opportunities to prosper; illegitimate political, administrative etc. pressure (*Kazmi, Amjad & Khan, 2008*). Questionnaires were administered directly to the subjects after their consent. Data collection was done between March and July 2013 from hospitals and private clinics in 8 major cities of the province Punjab.

## Mental wellbeing scale

The Warwick Edinburg Mental Wellbeing Scale (WEMWBS) was used to assess mental wellbeing; a reliable and validated scale (*Clarke et al., 2011*) which is well received in the English speaking Pakistani population (*Taggart et al., 2013*). WEMWBS showed good psychometric properties, and was cross culturally validated for Pakistani population. It exhibited excellent psychometric properties: good face validity, a unidimensional construct, a high internal consistency (0.89) and test–retest reliability and easy readability of WEMBS among Pakistani HCPs. Both principal component analysis and principle axis factoring demonstrated similar factor structure of WEMWBS (*Waqas et al., 2015*). It contains 14-positive items covering both eudaimonic and hedonic aspects of mental wellbeing and is scored on a 1–5 Likert scale to assess the frequency of occurrence of each item, with 1 corresponding to "None of the time" and 5 corresponding to "All of the time" in an ordinal fashion. Total score ranges from 14 to 70, with higher scores representing higher levels of mental wellbeing.

## Dietary patterns

Intake of 5 major food groups (Table S1) was asked over past 7 days and reported as 1 day average to compare with USDA Dietary Guidelines-2010. Food items were analysed using a standardised conversion scale (Table S1) to convert the amount of food eaten into servings.

## Exercise

Exercise was recorded by a 7-day recall and presented in minutes. Exercise was divided into moderate, vigorous and muscle strengthening activities. American Heart Association guidelines for exercise were used (150 mins/week of moderate activity or 75 mins/week

**Table 1  Demographics of the participants ($N = 1,190$).**

| Healthcare profession | Frequency | Percentage (%) |
|---|---|---|
| Doctor | 837 | 70.3 |
| Nurse | 218 | 18.3 |
| Dentist | 135 | 11.3 |
| **Gender** | | |
| Male | 491 | 41.3 |
| Female | 699 | 58.7 |
| **Marital status** | | |
| Married | 678 | 57.0 |
| Single | 512 | 43.0 |

of vigorous activity or a combination of both) (*Haskell et al., 2007*). For the simplicity of analysis and interpretation, vigorous activity minutes were multiplied by 2. This helped in making the cut-offs for combination of the two activities i.e., individuals doing both moderate and vigorous activities. Recommended activity was set at when collective sum for two activities was equal to or more than 150 min/week.

## Statistical analysis

SPSS Inc., (Chicago, Illinois, USA) version 21 software was used for analysis Descriptive statistics are shown for gender, medical profession etc. One way ANOVA and Independent-sample *t*-test were used for the comparison of means. Multiple Linear Regression analysis was run to analyze association of scores on WEMWBS (dependent variable) with demographic characteristics of respondents, age, income, profession, psychosocial stressors and dietary habits entered as predictor variables. Histogram and P–P plots were visualized to assess the assumption of normality of data. Durbin Watson test, case wise diagnostics and values of variance inflation factor (VIF) and tolerance (TOL) were run to ensure all assumptions of multiple regression analysis were met. Variables with more than two categories such as body mass index (BMI), profession of healthcare professionals and dietary habits were coded as dummy variables for multiple regression analysis. *P* values $<.05$ were considered statistically significant.

## Ethics statement

CMH Lahore Medical and Dental College Ethical Review Committee approved the study questionnaire.

## RESULTS

Large proportion of the final sample was occupied by Doctors, as shown in Table 1, with 58.7% females and 41.3% males. Mean working hours/week for Doctors $= 64.98 \pm 16.86$ S.D with junior doctors working an average of 7 h/week more than senior doctors (Table 2), Nurses $= 52.73 \pm 17.29$ S.D, Dentist $= 47.32 \pm 14.02$ S.D. When asked about the comfortable working hours, professionals from all groups demanded reduction in current working hours by 20, 12, 11 h respectively ($p < 0.05$). 50% of the participants were sleeping $<7$ h/day with doctors leading the list (74.5%). Doctors (66.9%) also lead the
**Table 2  A comparison of junior doctors and senior doctors.**

| Variable | Junior doctors [Mean (S.D)] | Senior doctors [Mean (S.D)] | *p*-value |
|---|---|---|---|
| $N = 837$ | 545 | 292 | – |
| WEMWBS score[a] | 46.76 (9.63) | 49.52 (9.22) | .003 |
| Breakfast/Week[a] | 4.76 (2.30) | 5.84 (1.95) | .001 |
| Fast-food/Week[a] | 2.70 (2.05) | 1.46 (1.76) | .001 |
| Servings of Fruits/Day[a] | 2.54 (2.35) | 2.21 (1.72) | .001 |
| Servings of Vegetables/Day[a] | 1.90 (1.45) | 2.18 (1.36) | .002 |
| Working hours/Week[a] | 67.33 (17.13) | 60.58 (15.43) | .001 |
| Do exercise[b] | 18.2% | 28.4% | .002 |
| Would change their profession if had the option[b] | 30.3% | 24% | .02 |

**Notes.**

[a] Independent sample *t*-test.
[b] Pearson chi-square.

list of healthcare professionals getting recommended sleep (7–8 h)(*Foley et al., 2004*). The most common chronic disease was visual defects (22.6%) like Myopia or Hyperopia next to back problems (9.7%) (Table 3). 10.4% of the participants were identified as smokers, smoking a median of 8 cigarettes/day. Only 34.5% were spending as much time with their family as much they like to. 70.3% of the participants (F > M) perceived themselves to be under occupational stress (Table 4). A significant majority (71.6%) said that they would not change their profession if they had the option.

## Junior vs. senior doctor

Junior doctors, as opposed to senior doctors, had lower WEMWBS scores, lower breakfast frequency, lower vegetable intake, high fruit intake, higher working hours and a relatively potent desire to change their profession (Table 2).

## Diet

Out of 1,190 participants, only one healthcare professional was taking the recommended servings. Protein over-eating was consistently high in all the 3 groups of healthcare professionals while the most neglected groups were Vegetables, Dairy and then Fruits (Table 3). Grains group had most professionals (60.6%) taking the recommended servings. 49.6% of the participants ate breakfast daily. 71.9% of the participants ate fast-food less than 5 times/week. Most of participants (58.2%) were at a healthy weight (F > M); 29.5% were in pre-obesity range (M > F); 7.3% were under-weight (F > M); 5% were obese (M > F). About 34.5% drank <6 glasses of water in a single day. Tea was the most frequently consumed (83.3%) caffeinated drink while only 25.7% of the participants consumed coffee; both with a frequency of 1–2 times a day.

## Exercise

76.2% of healthcare professionals did not exercise at all (F > M) with doctors making the largest contribution (Table 3). Out of those who did exercise, only 12.5% did

**Table 3** Diet and physical health related characteristics of healthcare professionals.

| | | Doctor | Nurse | Dentist | Total |
|---|---|---|---|---|---|
| Carbohydrates (6–11)[a] | Rec. | 58.3% | 72.0% | 56.3% | 60.6% |
| | <Rec. | 34.5% | 24.3% | 36.3% | 32.9% |
| | >Rec. | 7.2% | 3.7% | 7.4% | 6.6% |
| Proteins (5–6 1/2)[a] | Rec. | 21.5% | 12.4% | 20.7% | 19.7% |
| | <Rec. | 34.6% | 70.6% | 41.5% | 42.0% |
| | >Rec. | 43.8% | 17.0% | 37.8% | 38.2% |
| Dairy (2–3)[a] | Rec. | 8.2% | 6.4% | 7.4% | 7.8% |
| | <Rec. | 89.2% | 89.4% | 91.1% | 89.5% |
| | >Rec. | 2.5% | 4.1% | 1.5% | 2.7% |
| Fruit (4–6)[a] | Rec. | 9.0% | 4.6% | 8.1% | 8.1% |
| | <Rec. | 86.6% | 91.3% | 87.4% | 87.6% |
| | >Rec. | 4.4% | 4.1% | 4.4% | 4.4% |
| Vegetables (4–6)[a] | Rec. | 3.8% | 6.9% | 3.7% | 4.4% |
| | <Rec. | 94.3% | 91.3% | 93.3% | 93.6% |
| | >Rec. | 1.9% | 1.8% | 3.0% | 2.0% |
| Fat (saturated) (Infrequently)[a] | Infrequently | 57.8% | 68.3% | 37.8% | 57.5% |
| | Often | 34.3% | 26.6% | 43.7% | 33.9% |
| | Frequently | 7.9% | 5.0% | 18.5% | 8.6% |
| Fat (unsaturated) (Infrequently)[a] | Infrequently | 68.5% | 70.6% | 42.2% | 65.9% |
| | Often | 26.5% | 26.1% | 43.0% | 28.3% |
| | Frequently | 5.0% | 3.2% | 14.8% | 5.8% |
| Sugars (Infrequently)[a] | Infrequently | 29.0% | 39.9% | 32.6% | 31.4% |
| | Often | 40.9% | 42.7% | 40.0% | 41.1% |
| | Frequently | 30.1% | 17.4% | 27.4% | 27.5% |
| Water glass(250ml)/Day | 1<6 | 35.2% | 20.2% | 53.3% | 34.5% |
| | 6-8 | 36.2% | 43.1% | 26.7% | 36.4% |
| | >8 | 28.6% | 36.7% | 20.0% | 29.1% |
| Breakfast frequency/Week | Daily | 47.9% | 55.0% | 51.1% | 49.6% |
| | <7 | 47.2% | 41.3% | 44.4% | 45.8% |
| | None | 4.9% | 3.7% | 4.4% | 4.6% |
| Fast food frequency/Week | >7 | 1.7% | 0.9% | 0.0% | 1.3% |
| | 5–7 | 10.5% | 5.5% | 14.8% | 10.1% |
| | <5 | 73.6% | 62.4% | 77.0% | 71.9% |
| | None | 14.2% | 31.2% | 8.1% | 16.6% |
| BMI[b] (kg/m$^2$) | <18.5 | 7.4% | 6.9% | 7.4% | 7.3% |
| | 18.5 ⩽ 24.9 | 61.2% | 45.9% | 60.0% | 58.2% |
| | 25 ⩽ 30 | 26.8% | 39.4% | 30.4% | 29.5% |
| | >30 | 4.7% | 7.8% | 2.2% | 5.0% |
| Exercise[c] | ⩾ Rec. | 11.4% | 7.8% | 27.4% | 12.5% |
| | <Rec. | 10.4% | 15.1% | 10.4% | 11.3% |
| | None | 78.3% | 77.1% | 62.2% | 76.2% |

Table 3 (*continued*)

|  |  | Doctor | Nurse | Dentist | Total |
|---|---|---|---|---|---|
| Chronic diseases | Coronary heart disease | 1.9% | 3.2% | 0.7% | 2.0% |
|  | Hypertension | 7.3% | 8.3% | 5.2% | 7.2% |
|  | Diabetes mellitus | 4.2% | 6.0% | 3.0% | 4.4% |
|  | Visual defects | 27.0% | 10.1% | 15.6% | 22.6% |
|  | Back problems | 9.1% | 13.3% | 8.1% | 9.7% |

**Notes.**
[a] Servings/Day according to DIETARY GUIDELINES FOR AMERICANS, 2010 (between 1,800–2,600 Calories).
[b] Body-mass index (BMI) cut-off points by WHO.
[c] Exercise guidelines by American Heart Association (150 mins/week of moderate activity or 75 mins/week of vigorous activity or a combination of both).

**Table 4 Perception of occupational stressors by healthcare professionals.**

| Occupational stressors | Doctor | Nurse | Dentist | Total |
|---|---|---|---|---|
| Long working hours | 59.9% | 34.9% | 49.6% | 54.1% |
| Patient overload | 62.8% | 42.2% | 49.6% | 57.6% |
| Uncertain future | 56.9% | 28.0% | 49.6% | 50.8% |
| Insufficient opportunities to prosper | 56.3% | 26.6% | 47.4% | 49.8% |
| Illegitimate political, administrative, etc. pressure | 50.7% | 22.0% | 40.0% | 44.2% |

recommended exercise (M > F) with dentists making the largest contribution. Only 3.52% of the participants did any muscle strengthening exercise.

## Mental wellbeing

Mean WEMWBS score of the entire sample was $47.97 \pm 9.53$ S.D. Doctors scored $47.72 \pm 9.57$ S.D, Nurses $47.73 \pm 9.44$ S.D and dentists scored $49.92 \pm 9.26$ S.D. Dentists showed the highest levels of metal wellbeing as compared to doctors and nurses ($p < 0.05$). Male participants scored higher ($49.07 \pm 9.28$ S.D) than their female counterparts ($47.21 \pm 9.64$ S.D) ($p < 0.05$).

All assumptions of multiple regression analysis were met (Table 5). Multiple regression analysis revealed that mental wellbeing is positively associated with having breakfast daily, supplement intake, unsaturated fatty acids often instead of very frequently, presence of hypertension, working recommended and > recommended hours (7–8 h) and comparing with low intake of grains, proteins and fruits according to USDA guidelines; HCPs who reported high grain intake, high protein intake and high fruit diet had higher WEMWBS scores. Whereas eating restaurant made meals, high coffee and tea intake, back problems, low BMI, perception of not being treated properly in the society, desire to choose a different profession, perceived occupational stress, perceiving administrative and political pressure, doing no exercise compared to both AHA recommended or less than AHA recommended exercise regimes and low income were negatively associated with WEMWBS scores.

**Table 5** Multiple regression analysis for WEMWBS scores in healthcare professionals ($n = 1{,}190$).

| Variable | Under standardized coefficients | | Standardized coefficients | | 95% CI for B | |
|---|---|---|---|---|---|---|
| | B | Std. error B | Beta | *p*-value | Lower bound | Upper bound |
| (Constant) | 52.150 | 3.360 | | .000 | 45.558 | 58.742 |
| Most of the meals you eat are restaurant made? | −1.150 | .627 | −.049 | .067 | −2.379 | .080 |
| How many cups of tea you take in a day? | −.377 | .137 | −.072 | .006 | −.645 | −.109 |
| How many cups of coffee you take in a day? | −.563 | .271 | −.054 | .038 | −1.095 | −.031 |
| Do you take any supplements (Iron or Vit.D etc.)? | 1.336 | .638 | .054 | .037 | .084 | 2.588 |
| Hypertension | 2.059 | .958 | .056 | .032 | .180 | 3.939 |
| Back problems | −2.762 | .829 | −.086 | .001 | −4.389 | −1.135 |
| Do you think you are treated in a way you deserve to be treated in the society? | −2.885 | .543 | −.146 | .000 | −3.951 | −1.819 |
| If you could go back in time, would you choose a different profession? | 3.136 | .552 | .148 | .000 | 2.053 | 4.220 |
| Do you feel under occupational stress? | −4.081 | .743 | −.196 | .000 | −5.539 | −2.622 |
| Uncertain future | 2.639 | .816 | .138 | .001 | 1.038 | 4.239 |
| Illegitimate political, administrative pressure | −2.009 | .756 | −.105 | .008 | −3.492 | −.526 |
| Unsaturated fatty acid intake | 1.139 | .526 | .057 | .031 | .107 | 2.171 |
| Exercise (recommended vs. none) | −2.500 | .752 | −.087 | .001 | −3.975 | −1.025 |
| Exercise (<recommended vs. none) | −1.684 | .785 | −.056 | .032 | −3.225 | −.143 |
| Grain intake | 1.331 | .529 | .066 | .012 | .293 | 2.368 |
| Protein intake | 1.086 | .516 | .056 | .036 | .073 | 2.099 |
| Fruit intake | 1.681 | .761 | .058 | .027 | .189 | 3.174 |
| Body mass index | 2.191 | .944 | .060 | .020 | .339 | 4.043 |
| Working hours | 3.490 | .745 | .128 | .000 | 2.029 | 4.951 |
| Breakfast intake | 1.215 | .506 | .064 | .017 | .222 | 2.209 |
| Income | −2.120 | .846 | −.068 | .012 | −3.781 | −.460 |

**Notes.**
Method: backward, Adjusted R square = 23.6%, ANOVA $P < .001$.

# DISCUSSION

Our study showed that healthcare professionals do not "practice what they preach." They tend to be smokers, work excessively, have unhealthy dietary patterns and do not exercise or sleep according to recommendations. Dietary pattern of healthcare professionals of Pakistan are typical of ordinary citizens which is dominated by meat consumption and is low in fruits and vegetables (*Gallup Pakistan, 2011*). About 50% of Canadian physicians eat the recommended amount of fruit and vegetables (*Frank & Segura, 2009*) while only 9% of Pakistani physicians eat recommended serving of fruit and 3.8% eat recommended servings of vegetables. Despite poor dietary habits, 61.2% of Pakistani doctors [F > M ($p < 0.05$)] have a healthy weight as opposed to 54% of Canadian doctors (F > M), 26.8% Pakistani doctors are over-weight as opposed to 37% of Canadian doctors. 4.7% of Pakistani doctors are obese as opposed to 8% of Canadian doctors and 7.4% of Pakistani doctors are under-weight as opposed to 1% of Canadian doctors (*Frank & Segura, 2009*). These findings are paradoxical if we look at the dietary and exercise habits of Pakistani

healthcare professionals. These findings might be due to differences in total calorie intake, possibly larger portion sizes, consumption of high calorie snacks such as sugary drinks, cakes and biscuits etc. between meals and also alcohol may be higher in the Canadian doctors. There is also a difference in ethnicity, body fat distribution for the same BMI etc. which has led to the debate about setting different BMI guidelines for Asian population (*WHO Expert Consultation, 2004*). More than 3/4th of healthcare professionals do not exercise with less than 13% doctors getting the recommended exercise as opposed to 21% doctors in UK (*Gupta & Fan, 2009*) while Canadian doctors exercise 20–25 min daily (*Frank & Segura, 2009*) less than the recommendations of American Heart Association (AHA) (*Haskell et al., 2007*). Mental well-being scores of healthcare professionals are at an intermediate level which corresponds to an intermediate state of mental wellbeing. Professionals with higher incomes scored better because financial security gives you peace of mind and aids in achieving good health (*Marmot, 2002*). Healthcare professionals who do not eat breakfast daily either due to shortage of time or not being in the habit scored lower on WEMWBS which shows that a lifestyle without daily breakfasts renders you less mentally resilient. Females scored lower on WEMWBS than males possibly due to their susceptibility towards stress and its negative consequences more than their male counterparts (*American Psychological Association, 2010*). The incidence of CHD, Hypertension and Diabetes is less in healthcare professionals than the general population of Pakistan (*Pakistan Medical Research Council. Islamabad (Pakistan), 1998*). The injurious effects of smoking cigarettes are known to all. Because of all these effects, the number of people smoking cigarettes is shrinking in developing countries but paradoxically the number is on the rise in Pakistan, which is reflected in 12.7% doctors smoking cigarettes as compared to 3.3% doctors in Canada and 4% doctors in America (*Frank & Segura, 2009*). Almost half of the healthcare professionals sleep <7 h which may be due to job stress, long working hours, etc. Healthcare professionals have to work, on average, >50 h/week to compensate for the scarcity of resources e.g., man power etc. (*Pakistan Medical and Dental Council (PMDC)*) and in turn take a heavy toll on their own health and family life. Lack of sufficient sleep and long working hours lead to increased mistakes and hospital mortality which is especially a cause of concern in a country which has very limited number of healthcare professionals to share the heavy patient load (*Weinger & Ancoli-Israel, 2002*; *Editorial, 2009*). Doctors work almost 17 h/week more than the European Work Time Directive (EWTD) (*The Europen Parliament, 2003*) which limits the working of doctors to 40–48 h/week. No such directive is present at the moment in Pakistan which in a way opens the flood gates for doctors and other healthcare professionals to suffer from the ill effects of long working hours e.g., burnout, drowsiness during driving, mistakes etc. This also affects their family life, with only 34.5% healthcare professionals being able to spend time with their families as much as they would prefer. Junior doctors, when compared with senior doctors had lower mental wellbeing scores, lower breakfast frequency, greater fast-food frequency, longer working hours, lower exercise and poor job satisfaction (Table 2). Occupational stress is part of any job but it is of paramount importance in professions dealing with lives (*National Institute of Occupational Safety and Health, 1988*).

Healthcare professionals in Jamaica reported work related stress (occupational stress) to be 4 times more than non-work related stress (*Lindo et al., 2006*). 70% of Pakistani healthcare professionals said they perceive themselves to be under occupational stress which works as a double edged sword damaging the wellbeing of healthcare professionals at one end and causing poor patient care at the other. Despite all the difficulties in healthcare profession only 28% healthcare professionals would change their profession, if given a chance, which shows sheer dedication and resilience towards patient welfare and stressors, respectively.

## Limitations and recommendations for further research

- The proportion of dentists was small because of severe shortage of such professionals in the country e.g., there is only 1 dentist for a population of 12,000 (*Pakistan Medical and Dental Council (PMDC)*) as opposed to 1 dentist for a population of 1,600 in America (*The Henry J. Kaiser Family Foundation*) and UK (*General Dental Council UK*; *The World Bank, 2014*).
- Salt intake was not enquired about.
- Non-probability sampling method was employed for data collection.
- USDA Dietary Guidelines-2010 were used since there are no such dietary guidelines available for Pakistani population.
- Occupational stress was not ascertained by using a scale to prevent the questionnaire from being too long. Instead subjective perception of the presence or absence of it was enquired about. Future research should follow a more objective approach.
- BMI was calculated by using the values of height and weight given by the subjects instead of using any measuring instrument e.g., weighing machine.
- Further research should be initiated to see if there is a need of separate BMI guidelines for Asian population.
- Most of the healthcare professionals did not have any knowledge of serving size which made quantification of food items difficult. This problem was identified in the pilot and was later solved by using standardised "Food Exchange Lists" (Table S1).

### Funding
The authors received no funding for this work.

### Competing Interests
The authors declare there are no competing interests.

### Author Contributions
- Waqas Ahmad and Muhammad Shoaib Shafique conceived and designed the experiments, performed the experiments, analyzed the data, contributed reagents/materials/analysis tools, wrote the paper, prepared figures and/or tables, reviewed drafts of the paper.

- Frances Taggart conceived and designed the experiments, analyzed the data, contributed reagents/materials/analysis tools, wrote the paper, prepared figures and/or tables, reviewed drafts of the paper.
- Yumna Muzafar performed the experiments, analyzed the data, contributed reagents/materials/analysis tools, wrote the paper, prepared figures and/or tables, reviewed drafts of the paper.
- Shehnam Abidi, Noor Ghani, Zahra Malik, Tehmina Zahid and Naila Ghaffar performed the experiments, analyzed the data, wrote the paper, prepared figures and/or tables, reviewed drafts of the paper.
- Ahmed Waqas performed the experiments, analyzed the data, contributed reagents/materials/analysis tools, wrote the paper, reviewed drafts of the paper.

### Human Ethics

The following information was supplied relating to ethical approvals (i.e., approving body and any reference numbers):

CMH Lahore Medical and Dental College Ethical Review Committee approved the study questionnaire.

### Supplemental Information

Supplemental information for this article can be found online at http://dx.doi.org/10.7717/peerj.1250#supplemental-information.

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
