# Peer review of "Diet, exercise and mental-wellbeing of healthcare professionals (doctors, dentists and nurses) in Pakistan"

_PeerJ, doi:10.7717/peerj.1250_

## Round 0.1 · original submission · Major Revisions

Based on comments from reviewers, authors are advised to address their concerns especially with regards to objectives and data analyses. I would be happy to reconsider the manuscript after revision.

·

Basic reporting

Pass

Experimental design

Please refer to the general comment/attached document.

Validity of the findings

Pass

Additional comments

Title was appropriate.

Abstract was clear and comprehensible

Introduction
It was too brief and I love to know more about what is known and what is unknown about the topic. I think the research objective was very broad and vague which is very difficult to be handled by a cross-section study. In fact “practice what they preach” is also very subjective that is vulnerable to multiple interpretations.

"We initiated this study to see if healthcare professionals of the sixth most populous country in the world “practice what they preach”. (LINE 71-72)"

I recommend for the authors to rephrase the research objective to reflect the expected outcomes.

Methodology:
What was the purpose of conducting pilot study? Why merely focus on dietary intake?
Otherwise satisfactory

Results
Table 2 – I think the authors must include the values of t-statistics/X2-statistics for each comparison made. Otherwise the report is incomplete.

Table 4 – Occupational stressor: how the authors identified the stressors because none of this was mentioned in the methodology.

Table 5 – Binary logistic regression: what was the method used (enter method, stepwise method, etc)? What was the value of X2-statsitics for the whole model? What was the value of Nagelkerke R2? What was the value of Wald-X2 statistics for each variable in the model?

Overall the binary logistic regression looks okay, but the way it is presented can be improved.

Discussion
Overall the authors able to compare this study findings with previous studies. They also described about the limitations of this study – perhaps the non-probability sampling method could be added as another limitation.

Citation and reference
- I noted several references were not cited according to the journal format e.g.:
“They are more susceptible to stress and its 57 negative consequences than general population (Willcock et al., 2004),(P, DAVIDSON & N,58 2004)”. (LINE 56-58)

Overall
An interesting article that is publishable, but several amendments would help to improve the quality of this article.

·

Basic reporting

WEMWBS as an abbreviation of a scale was not introduced in abstract and in relevant part of the manuscript. It was only explained later in the text, after a number of references made to the abbreviation.

In text citation should be improved e.g. line 57.

Experimental design

In the introduction, the authors failed to specify clearly their objectives of this study. To see whether the health professionals do "practice what they preach" as an objective is a very vague scientific objective. Inability to specify the objectives truly affected the way the authors looked at the data and performed analysis.

In the methods, I would suggest that the authors can do away with the term "pilot study". In addition, WEMWBS scale was not explained adequately in term of its validity and reliability. For example, range of factor loadings, number of factors and response options were not explained. In addition, I am not in favour of determination of internal structure validity by Principal Component Analysis as described by the authors.

Stemming from the vagueness of the objectives, the statistical analyses as described by the authors were not specific. I would query the decision made by the authors to dichotomize WEMWBS scores into high and low categories. It defeated the purpose of obtaining the total score as a continuous numerical variable or at least very close to it. A lot of information contained in the numerical variable were lost by dichotomizing it into two categories. I would suggest that the authors repeat the analysis with multiple linear regression analysis. Another possible analysis is multifactorial ANOVA. The decision should be based on the objectives.

Validity of the findings

As mentioned before, the authors should have specified clear objectives, which would lead to selection of appropriate statistical analyses. I would like to withhold comments on the validity of the findings until I have clear views of the study.

---

## Round 0.2 · accepted · Accept

Responses made in the revised version are assessed to be adequate. It is now considered suitable for publication.